# Modeling the Electronic Absorption Spectra of the Indocarbocyanine Cy3

**DOI:** 10.3390/molecules27134062

**Published:** 2022-06-24

**Authors:** Mohammed I. Sorour, Andrew H. Marcus, Spiridoula Matsika

**Affiliations:** 1Department of Chemistry, Temple University, Philadelphia, PA 19122, USA; mohammed.sorour@temple.edu; 2Department of Chemistry and Biochemistry, University of Oregon, Eugene, OR 97403, USA; ahmarcus@uoregon.edu

**Keywords:** cyanines, Cy3, optical spectra, ensemble approaches, QM/MM, molecular dynamics, Wigner distribution, Franck–Condon, FC, ab initio, adiabatic Hessian FC

## Abstract

Accurate modeling of optical spectra requires careful treatment of the molecular structures and vibronic, environmental, and thermal contributions. The accuracy of the computational methods used to simulate absorption spectra is limited by their ability to account for all the factors that affect the spectral shapes and energetics. The ensemble-based approaches are widely used to model the absorption spectra of molecules in the condensed-phase, and their performance is system dependent. The Franck–Condon approach is suitable for simulating high resolution spectra of rigid systems, and its accuracy is limited mainly by the harmonic approximation. In this work, the absorption spectrum of the widely used cyanine Cy3 is simulated using the ensemble approach via classical and quantum sampling, as well as, the Franck–Condon approach. The factors limiting the ensemble approaches, including the sampling and force field effects, are tested, while the vertical and adiabatic harmonic approximations of the Franck–Condon approach are also systematically examined. Our results show that all the vertical methods, including the ensemble approach, are not suitable to model the absorption spectrum of Cy3, and recommend the adiabatic methods as suitable approaches for the modeling of spectra with strong vibronic contributions. We find that the thermal effects, the low frequency modes, and the simultaneous vibrational excitations have prominent contributions to the Cy3 spectrum. The inclusion of the solvent stabilizes the energetics significantly, while its negligible effect on the spectral shapes aligns well with the experimental observations.

## 1. Introduction

The Cy3 dye, 1,1’-dimethyl-3,3,3’,3’-tetramethylindocarbocyanine (see Figure 1), is one of the most commonly used indocarbocyanine probes in optical experiments on proteins and DNA [1,2,3,4]. Accurate modeling of its spectral signatures is critical for the interpretation of such experiments. The Cy3 absorption spectrum has unique spectral features with distinctive vibrational signatures. The spectrum exhibits a vibrational progression with the first absorption band (0-0) centered around 18,200 cm${}^{-1}$ followed by two absorption bands, 0-1 and 0-2, with ∼1150 cm${}^{-1}$ spacing [5,6,7]. This vibrational progression is common in the cyanine dye family and is thought to be dominated by a single vibrational mode corresponding to a totally symmetric polymethine chain C-C stretching mode in the electronic excited state [7,8,9]. The exact vibrational mode responsible for the vibrational progression is not well-defined, and the concept of a single dominant vibrational mode responsible for the first spectral shoulder has also been debated in favor of multiple vibrational excitations [8,10,11]. Computationally, robust interpretation of experimental spectra is limited by the ability of the simulation methods to account for the factors that contribute to the molecular spectra. Modeling the spectral signatures of a relatively large cation such as Cy3 with good accuracy remains challenging due to the need for accurate computational methods that can correctly describe the potential energy surfaces and the electronic properties of the ground and excited states. It gets more complicated when the thermal and environmental contributions cannot be ignored. Cy3’s monomeric spectral features are not very sensitive to the surroundings; the shapes and energetics are almost identical in water, methanol, and DNA environments [5,6,12,13]. Another degree of complexity in modeling absorption spectra is the number of electronic states that contribute to certain spectral bands. Cy3’s UV spectrum is well-known to be due to a strongly allowed HOMO → LUMO ($\pi \pi^{*}$) transition [14], and hence only a single excited state is required in modeling, simplifying the process. This makes Cy3 a perfect system for testing the ability of different computational approaches to capture its spectral signatures, as one needs not to worry about the environment contribution and the effect of multiple electronic states.

There are several methods that can be used to model the spectra of chromophores in solution [15,16,17,18]. Calculating vertical excitations at equilibrium (VEE) is the simplest approach to simulate the spectra and is usually used for the initial benchmarking of the ability of quantum mechanical methods to capture the energetics of the excited states. In this approach, VEEs are calculated, and the spectra can be plotted by means of Gaussian or Lorentzian functions [18,19]. This approach is mainly limited to comparisons with the maximum of experimental spectra, it does not account for the zero-point energy (ZPE), and it cannot give information about the vibronic effects or any broadening, especially when the thermal contributions are prominent [18,19].

In the ensemble approach, nuclear configurations of the ground state are commonly sampled using classical molecular dynamics (MD) [15,16,20,21]. Vertical excitations are calculated using geometries collected along the simulation time after equilibration has been reached. This method allows the inclusion of the environment and temperature effects on the ground state conformations including the anharmonicity. Proper sampling should in principal represent the nuclear potential energy surface curvature including the anharmonic regions [18]. This approach is of particular importance to flexible and charged systems where the interactions with the surroundings and the temperature have a strong effect on the ground state structure [18]. The classical ensemble approach does not account for the ZPE which is much larger compared to the thermal energy effects from the classical dynamics, and it cannot account for the thermal population of the vibrational states [20,21]. Moreover, the pure classical treatment of the nuclei represents a big challenge for the sampling, as can be understood by comparing the classical probability distribution of a classical harmonic oscillator to the quantum probability distribution of the wavefunction in its zeroth vibrational level [18]. Such a challenge can be overcome by obtaining the ensemble from ab initio molecular dynamics methods where the ZPE is accounted for [20,22]. The latter approach is efficient for rigid and semirigid systems with reasonable size where short simulations are satisfactory. Quantum sampling through the Wigner distribution is another sampling approach that accounts for the ZPE effects in a very efficient way; it samples the ground state nuclear space by considering that the molecule is at its zero-point level [20]. Wigner distribution using the harmonic approximation is very efficient for sampling, and is often used with rigid molecules. Similar to the classical MD simulations, the geometries obtained from the ab initio MD simulations or Wigner distribution are used to calculate vertical excitations to obtain the absorption energies at many configurations, and the spectra can be obtained using a Gaussian summation over all the excitations [15,18,20]. A standard deviation is then chosen phenomenologically to generate a smooth spectrum.

The Franck–Condon (FC) method [23,24,25,26,27,28] is another approach to obtain the electronic absorption spectrum, and it uses the vibrational wavefunctions of the ground and excited states [19,28,29,30]. To simulate the spectra using certain FC approaches, the equilibrium geometries of the ground and excited states and their normal modes are required. The vibronic contributions to the spectrum are then considered by the nuclear wavefunctions of the respective states, and the intensity of the vibrational excitations is determined by the extent of overlap between the initial and final states [18,23,31]. Within the Born–Oppenheimer approximation, the total spectrum can be calculated as a sum of the spectra of each individual excitation (in the time-independent (TI) approach) [32]. On the other hand, the less computationally demanding time-dependent (TD) approach uses the Fourier transform of the dipole moment autocorrelation function [28,32,33]. The FC approximation uses the zeroth order term of the Taylor expansion of the electric transition dipole moment, and it is suitable when the dipole allowed transitions are the main contributors to the spectra, which is the case for Cy3. A more general scheme is the Franck–Condon/Herzberg–Teller (FCHT) approach, where the zeroth and first order terms of the Taylor expansion are computed [32,34]. The latter approach is necessary when the transition dipole magnitude of the allowed transition is not large and the higher order terms of the Taylor expansion will have more weights, as in the case of weakly allowed transitions [32]. Most of the implemented FC approaches calculate the vibrational wavefunctions of the initial and final states under the harmonic approximation [35,36,37]. The cartesian coordinates of the normal modes of the two electronic states are then related using the Duschinsky transformation which might be very sensitive to the harmonic approximation used [38,39]. Two major harmonic representations are used, the adiabatic and vertical representations [35,36,37,40]. The main distinction between these two representations is how the potential energy surface (PES) and frequencies of the final state are obtained/treated relative to the initial state [35,36]. In the adiabatic representation the PES of the final state is expanded around its minimum, termed adiabatic Hessian (AH) using the Santoro and coworkers’ terminology [35], while in the vertical it is expanded around the ground state minimum, termed the vertical Hessian (VH). Further approximations can be used. In the shift approaches, the PES description and the frequencies of the final state are taken to be the same as the ground state, while the PES is displaced either based on the geometry difference, in the adiabatic-shift (AS) scheme, or based on the energy gradient, in the vertical gradient (VG) scheme [29,35]. Several studies on the performance of the vertical and adiabatic harmonic approaches and how they affect the obtained vibronic spectra have been performed [39,41], and a very comprehensive comparison can be found in the work of Ferrer and Santoro [36]. The FC approach and its related methods that use the harmonic approximation are suitable for rigid and semi-rigid molecules where the temperature and the environment have limited effect on the conformation, and the anharmonicity does not have much contribution [18,31,42]. Despite the development of FC methods that account for the anharmonicity, it is still challenging for very flexible molecules where multiple conformations need to be taken into account [19,32,43]. Solvent effects are usually included via obtaining the equilibrium structures and frequencies in implicit solvation. Combined FC/ensemble approaches have been developed, although they are system dependent and computationally expensive [17,18].

In this contribution, we simulate the absorption spectrum of the cyanine dye Cy3 incorporated in an oligonucleotide or in solution, using both ensemble and FC approaches and comparing them. Classical sampling (MD) and quantum sampling (Wigner) are used in the ensemble approach, and temperature and explicit solvation effects are included by means of classical molecular dynamics and hybrid quantum mechanical molecular mechanical (QM/MM) calculations. Possible drawbacks of the AMBER-DYES force field are discussed and attempts to solve them are shown. Comparisons to experimental spectra in methanol and DNA environments are used to evaluate the different approaches. The performance of several TD-DFT functionals including long-range-corrected functionals was compared, and benchmarking against higher-level wavefunction methods were performed when necessary. The performance of the ensemble approaches is compared to the Franck–Condon approach. The Herzberg–Teller effects [34] on the spectra are probed using the FCHT approach. The comparisons are extended to the performance of the harmonic approximations where we compare the AH, AS, VH, and VG approaches [35]. To account for the temperature contribution, the calculations were run at zero and room temperature. The spectral shapes computed from the sum-over-states approach used in the TI method is studied against the path-integral approach used in the TD method.

## 2. Computational Methods

**Classical Ensemble:** The Cy3-DNA system shown in Figure 1 was built using Chimera [44] version 1.14. The system is adapted from the experimental work of Heussman et al. [5], but a shorter sequence was used to reduce the computational cost. The MD setup and simulation was handled by Gromacs/2018.8. [45]. The force field parameters of Cy3 were derived from Amber99SB-dyes force field [46]. The missing linker parameters were obtained from the general Amber force field [47] (GAFF). The electronic density matrix of Cy3 optimized with B3LYP/6-31G(d) and the Gaussian16 package [48] was used to obtain the restrained electrostatic potential [49] (RESP) charges. The RESP charges fitting was undertaken using the Antechamber [50] package. The DNA part of the system was treated by Amber99SB-ILDN force field [51,52]. The system was solvated with TIP3P water [53], 100 mM NaCl, 6 mM MgCl${}_{2}$, and neutralized by Na${}^{+}$ in a cubic box of length 84.90 Å, resulting in a total number of 60,432 atoms. The energy of the system was minimized in vacuo using a steepest descent algorithm for 1000 steps. For initial equilibration, the temperature was gradually increased to 298 K in an NVT ensemble for 1 ns using a velocity-rescaling thermostat [54,55]. Further equilibration to reach a constant pressure of 1 bar was run for 1 ns in an NPT ensemble using a Parrinello–Rahman barostat [56,57]. For the MD production, the system was propagated using the leapfrog integration scheme [58] for 200 ns in an NPT ensemble. The equilibrations and the MD-production with periodic boundary conditions were propagated using a step size of 2 fs. The cutoff distance for the non-bonded interactions was set to 15 Å, and the long-range electrostatic interactions were accounted for by the particle mesh Ewald summation method [59].

A total of 301 uncorrelated configurations that span the equilibrated time domain (∼150–200 ns) were sampled in three sets, the criteria used for determining the convergence and sampling is discussed in more details in the Appendix A. The vertical excitation calculations (QM/MM-MD) were calculated for the sampled geometries using a hybrid quantum mechanical molecular mechanical (QM/MM) method. The boundaries between the QM and MM regions were constructed by the link-atom scheme as implemented in the Chemshell suite V-3.5. [60]. The additive scheme was used for the energy calculations. The polarization of the QM region by the MM region was handled by the electrostatic embedding scheme [61]. Cy3 with an N-methyl group, as shown in Figure 1, constituted the QM region, and the rest of the system was represented by point charges from the respective force fields. Excited states in the QM region were calculated using the wavefunction-based methods, configuration-interaction singles ( CIS) [62], CIS with perturbative doubles, CIS(D) [63], second-order approximate coupled cluster singles and doubles (CC2) [64] and the algebraic diagrammatic construction scheme through second order (ADC(2)) [65]. The time-dependent density functional (TD-DFT) method was used as well, with the following functionals: B3LYP [66], CAM-B3LYP [67], LC-ωPBE [68], ωB97XD [69], M11 [70], and M06HF [71]. Turbomole [72] V6.5 was used for the CC2 and ADC(2) calculations, Gaussian16 [73] was used for CIS, CIS(D), and TD-DFT. The absorption spectrum is obtained from QM/MM-MD using summation over all the vertical excitations represented as stick spectra convoluted over Gaussian functions [18]. Spectral width of 250 cm${}^{-1}$ was used as it leads to smooth spectra.

**Quantum Ensemble:** A Wigner distribution within the harmonic approximation, as implemented in Newton-X V-2.2. [74], was used for the quantum sampling. In total, 200 configurations were generated from a Cy3 molecule with N-methyl group optimized with ωB97XD/Def2SVP in implicit solvent (methanol) via polarizable continuum model (PCM). Methanol was used as an implicit solvent due to its negligible effect on Cy3’s spectra when compared to DNA [5,75]. For the vertical excitations, Newton-X was paired with Turbomole [72] V6.5 for the ADC(2) calculations and it was paired with Gaussian09 [76] for the TD-DFT calculations. The TD-DFT vertical excitations were performed in implicit solvent (methanol). ADC(2) vertical excitations were undertaken in a vacuum because there is no implicit solvation in the Turbomole version we have, and as we will discuss later, the solvent does not seem to have much of an effect. The Wigner absorption spectrum is obtained using summation over all the vertical excitations convoluted with Gaussian functions, and smooth spectra were obtained using spectral width of 500 cm${}^{-1}$ [18].

**Franck–Condon Calculations:** All the Franck–Condon calculations were run using Gaussian16 [73]. The ground and excited state optimizations and frequency calculations were obtained in the gas-phase as well as in implicit solvent (methanol) via PCM. The performance of the harmonic approximations, adiabatic Hessian, adiabatic shift, vertical Hessian, and vertical gradient, were tested. Since we are only considering the first excited state which is known to be a strongly allowed transition, the use of the FCHT [77] approximation is not necessary. Hence, through the study, we only use the FC approximation where the electric transition dipole moment is truncated at the zeroth-order of the Taylor expansion. For completeness, FCHT is compared to FC in the Appendix A, where it is confirmed that it is not important. The absorption spectrum is obtained using the time-independent and the time-dependent approaches. The Gaussian16 package’s default spectral width of 150 cm${}^{-1}$ at half maximum was used for all the FC calculations, and it resulted in a satisfactory broadening. Lastly, all the optimization calculations in the ground and excited state through the whole study were performed using the equilibrium linear response solvation formalism [78,79].

## 3. Results and Discussion

In this section, the ability of the ensemble and Franck–Condon approaches to simulate the experimental absorption spectrum of Cy3 [5] is compared. The section is arranged by presenting: (1) the classical ensemble approach represented by the QM/MM-MD calculations, (2) the quantum ensemble approach represented by the Wigner distribution, (3) the Franck–Condon approach, and (4) comparison between the three approaches.

### 3.1. Classical Ensemble: QM/MM-MD

Classical molecular dynamics is a very convenient approach to simulate the complicated dynamics of huge heterogeneous systems. Provided that an accurate force field is used and proper sampling is performed, MD can provide an ensemble of geometries that their population distribution is a good representation of the ground state’s potential energy surface [18,20]. The results depend on several factors that are discussed here.

#### 3.1.1. Electronic Structure Effect

We first examine how the choice of electronic structure theory affects the absorption spectrum. In addition to the importance of the force field and the sampling to obtain the ensemble, the electronic structure method used for the excited states calculations has a very important role. The oscillator strength (f) used to obtain the spectra depends on the transition energy and the transition dipole moment. Hence, an electronic structure method that can give a consistent and accurate description of the excited state energies and properties is necessary. Cost effective methods are also desirable, not only because of the comparatively large size of the chromophore, but also because of the size of the ensemble that is usually hundreds of configurations [18,31,42]. When it comes to cost effectiveness, TD-DFT is a very efficient approach. It has been shown before that TD-DFT fails to accurately predict the excited state in cyanines [75,80,81,82,83], although its ability to predict the experimental transition dipole moment is still reasonable [75]. Since we are interested in the spectral shapes, we will test the performance of several functionals and compare it against wavefunction-based methods.

Figure 2 shows the spectra obtained from all the electronic structure methods used in this work, and it is clear that the choice of method has a prominent effect on the spectral energies, intensities, and shapes. The spectra obtained from the TD-DFT calculations are similar in shape and intensities for all functionals studied, and they are all lacking the distinctive vibrational progression peaks of the experimental spectrum. Similar behavior has been reported for a closely related set of functionals in a similar study by De-Vetta et al. [21], in which they simulated the absorption spectrum of Temoporfin. Similar to what De-Vetta et al. observed, B3LYP displays the best performance when it comes to the energetics. For the purpose of our study on Cy3, the performance of all TD-DFT functionals is not satisfactory, and more investigations along with comparisons to wavefunction-based methods are necessary.

Turning to wavefunction methods, it is instructive to first compare CIS and CIS(D). CIS(D) has a perturbation correction to the CIS energies [63,84], yet the calculated excited state properties are the same as CIS, which means that the energies vary while the transition dipole moments are identical. A quick look at Figure 2a shows that the spectrum obtained from CIS is significantly different from the spectrum obtained from CIS(D). The spectrum obtained from CIS is narrower with no spectral features. This interesting behavior means that the inconsistency is originating from variations in the calculated excitation energies. This behavior motivated us to compare the performance of the more accurate methods CC2 and ADC(2). CC2 and ADC(2) are known to have close performance [85], and investigating their performance on this problem is a suitable benchmark. CC2 and ADC(2) predict some structure to the spectrum, compared to the other methods that are mostly structureless. Despite displaying significant improvement over all the other methods, however, the spectral shapes obtained from CC2 and ADC(2) fail to predict the experimental shape with good accuracy. As shown in Figure 2b and upon normalizing and shifting the spectra so that the maximum is matching the maximum of the experimental spectrum, CC2 and ADC(2) can predict the three main spectral bands but they fail to reproduce the vibrational spacing, the relative intensities, and the exact width. Moreover, there is an obvious shoulder at lower energies of the maximum that is not present in the experimental spectrum. The performance of ADC(2) is different from CC2, with CC2 agreeing better with the experiment in terms of the shape of the spectrum. At the ADC(2) level, the hump to the left of the maximum is more prominent, the relative intensities of the peaks are worse, and the spectrum is wider. A closer look on the energetics and the properties obtained from these two methods shows differences for the same geometries, and this is discussed in more detail in the Appendix A.

Figure 2a allows for comparison of the calculated intensities as well. The intensities vary widely for the different electronic structure methods. ADC(2) and CIS(D) show the lowest intensity maxima while CIS shows the maximum. The integrated intensities however are less variable. The spectra obtained from ADC(2) and CIS(D) are showing lower intensities but also wider peaks. Since the intensity ε is obtained from the oscillator strengths, we tried to see if there is any correlation between the ε maximum of the calculated spectra and the oscillator strength calculated at vertical excitation. This correlation plot (shown in Appendix A) shows some qualitative correlation between the two quantities, but not a great quantitative correlation. This further highlights that the properties at vertical excitation are not sufficient to describe the spectrum. Focusing further on intensities when comparing spectra and comparing the intensities to experimental ones is tricky, since ε depends on the parameters we choose to plot the spectrum, particularly σ. In Figure 2a all spectra use the same σ so a comparison is appropriate, but we will refrain from continuing such comparisons for other methods because of this limitation.

A most worrisome feature appears when we compare the spectrum obtained from the ADC(2) QM/MM-MD calculations to the gas phase spectrum using ADC(2) and the same Cy3 geometries. Figure 3 shows that the spectral features obtained from ADC(2) QM/MM-MD vanish when we run the excited state calculations in the gas phase (GS-MD). This behavior suggests that the vibrational progression obtained from ADC(2) is not emerging from the vibrations of the molecule itself, but is rather an effect from the environment included in the QM/MM calculations. This origin for the vibrational progression disagrees with what is known for this system, and with our following analysis.

#### 3.1.2. Sampling Effect

A statistically representative ensemble requires the generation of hundreds of configurations. Defining proper sampling criteria is quite challenging, and it varies with the purpose of the study and the nature of the problem. The larger the size of the sample the better, but there is no definitive limit. It gets more complicated when one needs to use electronic structure methods on medium and large size systems as the computational cost can increase rapidly. Sampling configurations from MD simulations requires sampling uncorrelated configurations from a well converged system. The convergence criteria and the sampling details are discussed in the Appendix A. For our system, we found that 150–200 configurations are enough to obtain a somewhat converged spectrum, although to reproduce the details more may be needed. Figure 4 shows that the electronic structure method has some effect on the convergence of the spectrum. CC2 shows smaller variations as the number of frames increases compared to ADC(2). The ωB97XD performance is fluctuating with an appearing/disappearing shoulder at the left of the spectrum. It is also important to note that the number of configurations does not have much effect on the width of the spectra.

#### 3.1.3. Force Field Effect

The accuracy of the force field used for molecular dynamics is the main determinant of the quality of the simulation. Classical force field parameters are obtained from, and for, systems in their ground state, and the classical dynamics cannot predict events related to the excited states. The spectral vibrational progression is usually manifested by an active vibrational motion coupled to the electronic transition. For cyanine dyes, a polymethine chain C-C stretching mode in the electronic excited state is thought to be responsible for the vibronic progression [7,8,9]. Comparing Cy3’s polymethine equilibrium bond lengths from Amber99SB-dyes force field [46] to the equilibrium structure obtained from ab initio methods shows a significant difference. As shown in Table 1, the classical force field leads to a complete loss of the polymethine chain bond delocalization. The localization imposed by the force field results in a series of alternating single and double bonds along with breakage of the structural symmetry of the molecule. In an attempt to solve this issue and guided by the structural parameters obtained from an ωB97XD/Def2SVP optimized Cy3, we adjusted the bond lengths and the corresponding force constants of the polymethine chain C-C bonds and the neighboring N-C bonds, as shown in Table 1.

Figure 5 shows the non-shifted spectra obtained from the unmodified [46] and modified force field using ωB97XD/Def2SVP-QM/MM. Adjusting the polymethine chain C-C bonds and the neighboring N-C bonds clearly has an effect on the spectrum resulting in a narrower and more symmetric spectrum. However, the attempted reparameterization did not reproduce the distinctive spectral features of the experimental spectrum. Nevertheless, it shows that it is very important when obtaining force field parameters to check whether they are appropriate for the system under investigation, since these parameters can have a significant effect on the spectra and other properties.

Other attempts to improve the parameters of the Amber99SB-dyes force field [46] were conducted by Shaw et al. [86], who used the analytical partial Hessian fitting approach [87] and the density-fitted [88] method rather than the HF level used by Graen et al. [46] to obtain the bonded parameters of Cy3 among other commonly used dyes. Shaw et al. implemented these parameters into their CHARMM-DYES force field [86]. These refined parameters did not solve the Cy3’s polymethine chain C-C bonds localization problem, and they gave equilibrium bond lengths similar, or even worse, to those obtained by Graen et al. [46], see Appendix A.

### 3.2. Quantum Ensemble: Wigner-Distribution

We now examine whether a quantum sampling can provide better results. Using the structure optimized at the ωB97XD/Def2SVP level in implicit methanol via PCM, the Wigner distribution was used to generate an ensemble of 200 configurations. Vertical excitation energies were calculated for the obtained configurations using ADC(2), B3LYP, CAM-B3LYP, and ωB97XD.

As shown in Figure 6, the spectra obtained from the Wigner ensemble fail to reproduce the experimental spectrum as well. The Wigner obtained ensemble leads to spectra that are wide, and they cannot capture the vibronic progression. The failure of the Wigner distribution might be attributed to the presence of anharmonicities that the harmonic approach cannot account for. Despite its ability to account for the ZPE of the ground state, possible changes to the potential energy surface of the target state and its normal mode frequencies are not taken into account. Similar to the rest of the ensemble-based vertical electronic excitation approaches, simultaneous vibrational excitations are not accounted for. The inability of the Wigner distribution to reproduce the spectral fine structure of molecules has been reported before by De-Vetta et al. [21].

### 3.3. Franck-Condon Approach

The Franck-Condon approach has been used to simulate the spectral signatures of several conjugated systems with good accuracy [89,90,91,92], and it allows for a quantum mechanical interpretation to the origin of the vibrational features of the cyanine chromophores [10,11]. Alia and Flack [11] applied the FC approach and an approach developed by their group on several cyanine dyes. In their work, the vibrationally-resolved spectrum of Cy3 was not calculated, although it was calculated for the structural analog Cy5 (1,1’- diethyl-3,3,3’,3’-tetramethylindodicarbocyanine). Based on the work of Alia and Flack [11], the FC TD approach at 298 K along with implicit solvation models gave reasonable spectra. They also found that B3LYP could produce accurate FC spectra. In this study, the FC absorption spectrum of Cy3 is calculated and the factors affecting the quality of the spectrum are investigated. In contrast to the good performance of B3LYP in predicting the energetics and spectrum of Cy5 [11,93], we found that this is not the case for Cy3. As we will show later, ωB97XD displays the best performance and we use it as the main TD-DFT functional for the Franck–Condon calculations.

#### 3.3.1. Spectra at 0 K

The adiabatic and vertical approaches are the main categories of the FC harmonic approximation, and the main difference between them is that the optimized structure and frequencies of the target state are not required to calculate the FC spectra in the vertical scheme [35,36,37]. For the vertical methods (VH and VG), an extrapolation of the final state displacements and vibrations from the initial state is used. This is suited to the very fast photo-excitation processes where events on the excited states can be neglected and is expected to give poor performance with the comparatively long lived events such as low frequency motions. On the other hand, the adiabatic approaches (AH and AS) take into account the time needed for the relaxation of the target state and the related events. Here we compare the performance of the adiabatic Hessian, adiabatic shift, vertical Hessian, and vertical gradient approaches using the time-independent scheme at 0 K. The ground and first excited state minimum structures and frequencies were obtained in the gas-phase (GS) using ωB97XD/Def2SVP.

As shown in Figure 7, the vertical approaches could not reproduce the second band around 19,500 cm${}^{-1}$ while the adiabatic approaches can do so successfully. The ability of the adiabatic approaches to predict the second band suggests the importance of structure and energetics of the target state and its long lived events. The adiabatic Hessian approach displays somewhat better performance compared to the adiabatic shift which predicts lower intensity for the second peak and wider peaks separation. The AH approach performs better than the other approaches energetically as well; the AH overestimated the energies at maximum absorbance ($\lambda_{max}$) by 4230 cm${}^{-1}$ compared to 4870 cm${}^{-1}$, or more, for the three other approaches. It should be highlighted that the energetic error is primarily because of the underlying electronic structure theory. The vertical excitation energy calculated at the ground state minimum is blue shifted by 4906 cm${}^{-1}$ compared to the experiment. We will use the more accurate AH approach for the upcoming tests starting with the origin of the vibrational progression.

#### 3.3.2. Origin of the Vibronic Structure

A very useful feature of the FC approach is that we can see which modes are contributing to the spectrum. Figure 8 shows some of the most important individual contributions as stick spectra. The normal modes 3, 30, 81, 100, 106, and 125 were found to be the modes with most contribution to the spectrum, and movies of these normal modes can be found in the Appendix A. Normal mode 3 makes the strongest contribution to the main band, and it is always involved in combination bands responsible for the second peak as well. This normal mode has a frequency of 34.9 cm${}^{-1}$ in the excited state, and it has a characteristic global bending of the backbone around the polymethine chain. A similar low frequency normal mode was found to have major contribution in the absorption spectrum of other cyanines and curcumin [10,11]. Some contributions to the main peak are originating from combination excitations to the normal modes 3 and 30, which give rise to a shoulder at higher energies of the band. This shoulder can be seen experimentally as well [6]. Mode number 30 has a frequency around 354 cm${}^{-1}$, and is characterized by a symmetric twisting of the dimethyl groups. The second main peak in the spectrum is dominated by combinations involving normal mode 3 along with other high frequency modes, mostly normal modes number 81, 100, 106, and 125. These modes have frequencies 1136, 1378, 1415, and 1525 cm${}^{-1}$, respectively, on the excited state, and they primarily involve motion around the polymethine chain. These results align well with the experimental picture [8] where the polymethine C-C stretching is thought to be responsible for the vibrational progression. They are however different from the simple idea of a single mode contributing to the vibrational progression of the spectrum. It is also interesting to highlight that all the high intensity peaks are combination bands with the low frequency mode 3 being involved. The results confirm the importance of the low frequency modes and the simultaneous excitations for the main peak and the first shoulder [10,11]. When compared to the vertical approaches, the importance of the low frequency modes becomes even more apparent. These models completely miss the second peak because they miss the contributions of mode 3 in the combination bands. In the vertical Hessian approach the spectrum is mainly built of fundamental bands while in the adiabatic Hessian only 1.5% of the bands are fundamental bands.

Even though the current results are far from the one-mode model often used to model the spectrum of Cy3 and other chromophores, it appears that the one-mode model can be a fair approximation because there is a ‘cluster’ of high frequency modes, (81, 100, 106 and 125), ranging from 1100–1500 cm${}^{-1}$, that dominate. Because these modes have frequencies that are relatively close together, they seem to merge into an apparent single mode with frequency about 1300 cm${}^{-1}$.

#### 3.3.3. Temperature Effect

In order to reproduce the experimental spectrum we need to consider the temperature effects as well. Here we test the temperature dependence with the time-independent and time-dependent schemes. The adiabatic Hessian approach is used with structures and frequencies obtained from ωB97XD/Def2SVP in the gas-phase. As shown in Figure 9, the performance of the TD approach is identical to the TI approach at 0 K, and that should be the same case when we use higher temperatures. The TI approach requires a very large number of summations of Franck–Condon integrals which is computationally expensive, especially at higher temperatures, and prescreening schemes have been developed to choose the most relevant transitions [39,94]. Using the prescreening scheme, we were able to get information about the most relevant vibrational transitions at room temperature but not much of an improvement to the spectral shape (see Appendix A). Relaxing the default prescreening criteria implemented in Gaussian16 [73], however, is computationally too expensive and cannot converge. On the other hand, the TD approach includes all the initial and final vibrational states at any temperatures at no extra computational cost which results in the calculation of more accurate spectral shapes [94,95].

The inclusion of temperature in the TD scheme leads to a significant improvement in the predicted spectrum; the TD/298 K calculation predicts the spectral width and relative peaks intensity with very good accuracy. Since the TD approach cannot give information about the vibrational transitions, the TI approach becomes very useful. The analysis of the most relevant transitions obtained from the TI/298 K reveal that thermal effects lead to the population of higher vibrational levels in the ground state which have prominent contribution to the vibronic excitations. We also see a significant contribution from combination vibrational excitations, similar to what we have seen in the TI/0 K analysis in Figure 8.

Our calculations predict that the separation between the 0-0 and 0-1 effective bands becomes somewhat smaller when temperature is included. Such an effect makes sense if we consider the population of low frequency modes in the ground state. However, it should be noted that there is no temperature effect in the experimental spectra in the range between 15${}^{\circ }$ and 85${}^{\circ }$ for the monomer Cy3 [6].

#### 3.3.4. Environment and Functional Effects

The simplest and most common way to include the environment effect in the FC approach is the relaxation of the ground and target states in implicit solvent. More expensive and sophisticated approaches that combine the ensemble and FC approaches have been developed [17,18], and they are very promising for systems where the environment contribution is non negligible. Since the DNA environment has limited to no effect on Cy3’s absorption spectrum, as seen by comparing the spectra in DNA, methanol and water [5,75], there is no need to use the complicated combined approaches. Hence here we will use methanol as an implicit solvent via PCM for the structural optimizations as a reasonable approximation. As shown in Figure 10a, the ωB97XD/FC/TD/298 K spectra obtained in gas-phase and in methanol PCM solvation suggest that the environment has a limited effect on the spectral shapes which aligns well with the experimental findings [5,12]. Despite the negligible effect on the spectral shapes, the solvent leads to a significant improvement in the predicted energetics. The spectrum obtained in methanol is blue shifted by 1820 cm${}^{-1}$ compared to 4230 cm${}^{-1}$ in gas-phase.

Previous studies have shown that the choice of the functional can affect the quality of the obtained FC spectra while the basis set choice has a negligible effect [11,89,93]. As shown in Figure 10, ωB97XD has a satisfactory consistent performance in predicting the spectral shapes, while B3LYP and especially CAM-B3LYP are less accurate. Using methanol as an implicit solvent, B3LYP showed a remarkable accuracy in predicting the position of the $\lambda_{max}$ with deviation of 280 cm${}^{-1}$ (0.035 eV). The dependence of the spectral features on the functional highlights the sensitivity of the spectrum on the ability of the electronic structure method to predict the minima and frequencies of both the ground and the excited state. This sensitivity of the vibronic spectra could potentially be used as another way to test the accuracy of methods in predicting excited state vibrational frequencies, which are difficult to benchmark.

### 3.4. Comparing the Ensemble and Franck–Condon Approaches

After discussing the different approaches, we will now compare the performance of the ensemble approaches to the FC approach. Due to its good performance in the FC approach, we choose to limit the comparison here to the results obtained from ωB97XD/Def2SVP. Comparisons of the performance of the other methods can be found in the Appendix A.

As shown in Figure 11, the FC approach drastically outperforms the ensemble approaches in predicting the spectral bands and shapes with good accuracy. Based on our analysis, the good performance of the FC approach using the adiabatic harmonic approximations is mainly because it takes into account (1) the differences between the ground and excited state equilibrium structures, energies, and frequencies, (2) the simultaneous electronic-vibrational excitations from the zeroth and higher ground vibrational levels, (3) the significant contributions from the low frequency modes that can be easily populated at room temperature, (4) the simultaneous vibrational excitations that we found to have significant contributions at finite temperature. Furthermore, the success of the FC model without needing the HT terms indicates that the effect of the change of the transition dipole moment for the different vibrational modes is negligible.

On the other hand, sampling configurations using the ensemble approaches is a substantial challenge. Quantum sampling takes into account the ZPE of the ground state, and the quality of the ensemble is highly dependent on the harmonic approximation used. Barbatti and Sen [20] showed that the quantum sampling through Wigner distribution is a superior ensemble generation approach to the ground classical and the Born–Oppenheimer ground-state dynamics, and it can deliver absolute width and heights. However, the vibrationally resolved signatures will be totally missed because the ensemble approach does not include any information about the target excited state [20]. Similar to the Wigner sampling, the classical sampling is not expected to capture the vibronic progression as it does not incorporate any information about the final state. Classical sampling is also highly sensitive to the quality of the force field, and as we have shown earlier, the force fields that we investigated for Cy3 [46,86,96] include structural parameters that deviate drastically from the quantum mechanical picture. In the current study, the Wigner distribution predicts a spectrum that is comparatively wide, while the classical sampling gives a very narrow spectrum. The inclusion of ZPE through the Wigner distribution does not improve the spectrum. It does give somewhat more structure to the spectrum but the relative intensities are completely different from the experimental spectrum and the overall shape and width are wrong. Another problematic behavior to highlight is that a structured spectrum can sometimes be predicted for the wrong reasons, as shown in Figure 3, where the ensemble approach erroneously attributed the observed structure in the spectrum to the solvent effects.

The Franck–Condon approach not only outperforms the ensemble approaches in predicting the spectral bands and shapes in good accuracy, it also displays a significant improvement in predicting the energetics, see Figure 11. The $\lambda_{max}$ predicted by the FC calculation overestimates the experimental maximum by 1820 cm${}^{-1}$ (0.23 eV) compared to 5200 cm${}^{-1}$ (0.64 eV) for the quantum sampling and 3750 cm${}^{-1}$ (0.46 eV) for the classical sampling. It is worth mentioning that the vertical electronic excitation at equilibrium overestimates the experimental maximum by 4906 cm${}^{-1}$ (0.60 eV). The significant difference between the FC energy at maximum and the energy of the vertical excitation at equilibrium confirms the unreliability of the VEE as a method to benchmark the performance of the computational methods against the experiments [11]. Part of the difference between the vertical excitation and the experimental maximum comes from the shape of the spectrum and not the electronic structure method. Due to the ability of the FC approach to provide reasonable spectral predictions, it constitutes a more reliable approach to benchmark the accuracy of the computational methods for systems with strong vibronic contributions, as shown in Figure 10b.

## 4. Conclusions

Accurate modeling of the spectral signatures of chromophores is of great importance to understanding their photophysics and photochemistry. Capturing and explaining the vibronic signatures is limited by the ability of the computational methods to account for the factors contributing to these features. In this paper, we tested the performance of the ensemble approach represented by the classical dynamics and quantum sampling via Wigner distribution, as well as the Franck–Condon approach and its harmonic approximations. The major finding is that all the vertical excitation based methods used in this work (FC/VH, FC/VG, VEE, Wigner, QM/MM-MD) fail to predict the spectral shape and the vibronic signature of the Cy3 absorption spectrum. The success of the adiabatic approaches (FC/AH, FC/AS) in capturing the vibronic bands and the spectral shape is clear evidence on the importance of the final state’s information for the fine structure of the spectrum of Cy3. The ensemble methods, both the QM/MM-MD and the Wigner distribution, fail completely to reproduce the vibronic progression, and they should not be used to model spectra with vibronic structure. If one is interested in the vibrational structure of the signal arriving from vibrations in the upper electronic potential, then the vertical excitation and the ensemble approaches that sample ground-state configurations only, are not suitable, and the Franck–Condon approach should be chosen. On the other hand, the Franck–Condon approach requires a well defined excited state minimum, so for molecules that do not have one it cannot be used. For that reason, the FC approach cannot be used to model the spectrum of bound-to-continuum transitions. Ensemble methods are also likely more appropriate in systems where electronic states rather than vibrational levels are responsible for the spectral shapes [97].

The Franck–Condon approach allows probing of the factors contributing to Cy3’s absorption spectrum. The most intense peaks are combination bands of a low frequency mode and high frequency modes that involve stretching of the polymethine C-C bonds. This assignment aligns well qualitatively with the experimental picture on the origin of the vibrational progression in the cyanine dyes, but there is considerably more complexity compared to the simple models of one effective mode contributing to the spectrum. Temperature has a prominent effect on the spectral shapes. On the other hand, the solvent has a negligible effect on the shape, while it has a prominent and important effect on the energetic position of the spectrum.

Although the current work provides insight into accurate representation of the spectrum of Cy3 and similar molecules, generalizing to provide suggestions about producing reliable spectra for any molecule is much harder. The problem of accurately simulating spectra has been discussed extensively in the literature, and we have summarized these points in the introduction and throughout the paper. Unfortunately, it is hard to suggest a general approach that is guaranteed to work. Some general guidelines about the choice of FC vs. ensemble approaches exist: small rigid molecules are better treated with the FC approach, while large flexible molecules which are expected to be sensitive to the environment and temperature are better treated with an ensemble approach. A combined FC/ensemble [18] approach has also been developed which combines these effects. However, it is hard to be certain a priori that any of these methods will produce an accurate spectrum. Choosing the correct electronic structure method is another component that has to be considered carefully and benchmarked. Benchmarking using similar molecules with existing experimental spectrum is always advisable.

A final conclusion from our work is that the most usual practice of benchmarking electronic structure methods by comparing vertical excitations to the experimental absorption maximum is problematic. When modeling the spectrum, the theoretical absorption maximum is different from the vertical excitation energy, and very much dependent on the theory used to model the spectrum. The oscillator strengths at equilibrium position are also not adequate to predict the experimental intensities. A proper comparison should be between equivalent features of the theory and experiment, ideally comparing the spectra themselves. This provides a benchmark not only of the excited state energies, but also of the frequencies of the states involved.

## Figures and Tables

**Figure 1 molecules-27-04062-f001:**
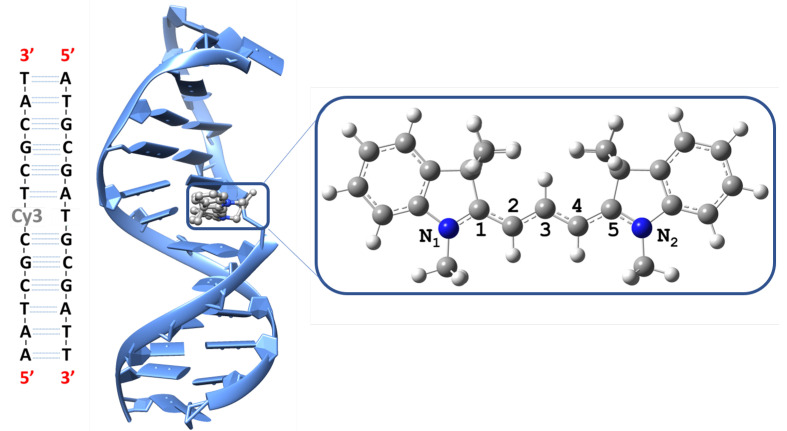
**Left**: The Cy3-DNA sequence studied [5]. **Middle**: Snapshot from the molecular dynamics equilibrated structure. **Right**: Cy3 ground state structure optimized using ωB97XD/Def2SVP showing the polymethine chain atoms numbered from 1 to 5.

**Figure 2 molecules-27-04062-f002:**
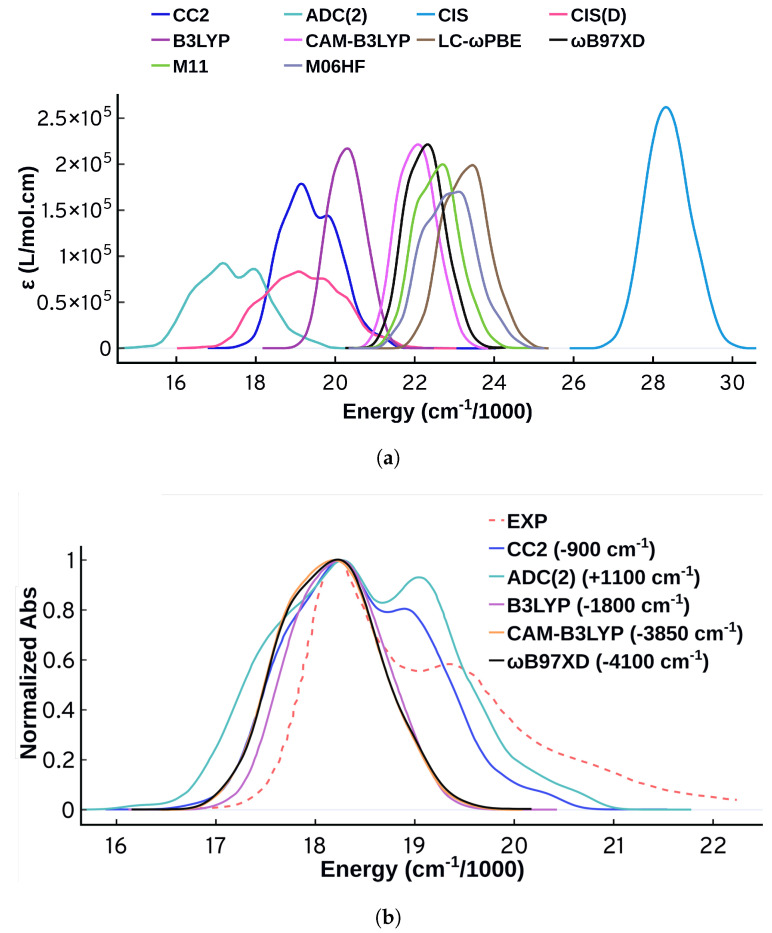
(**a**) Unnormalized absorption spectra from QM/MM-MD calculations using four wavefunction-based methods and six TD-DFT functionals using the same set of configurations (301 configurations). (**b**) Normalized QM/MM-MD absorption spectra shifted so that all the maxima are centered at the experimental maximum of 18,194 cm${}^{-1}$ (some methods are omitted for clarity). The shift direction and magnitude is included in parentheses. All the spectra were plotted using standard deviation of 250 cm${}^{-1}$. The experimental spectrum from the work of Heussman et al. [5] is also shown.

**Figure 3 molecules-27-04062-f003:**
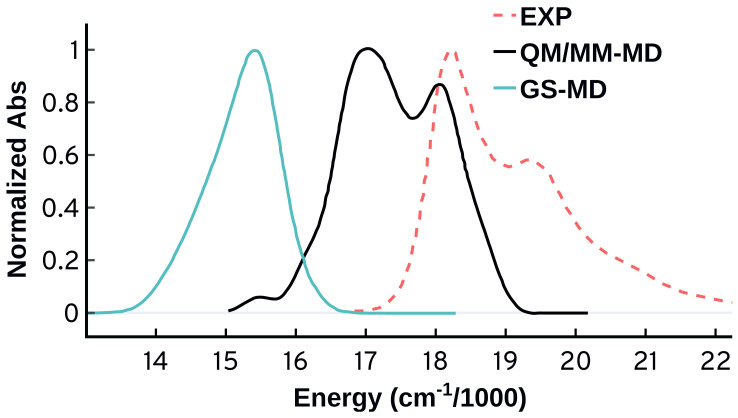
Normalized absorption spectra from the same MD ensemble (50 configurations) using the ADC(2) method in gas phase (GS-MD) and with QM/MM (QM/MM-MD).

**Figure 4 molecules-27-04062-f004:**
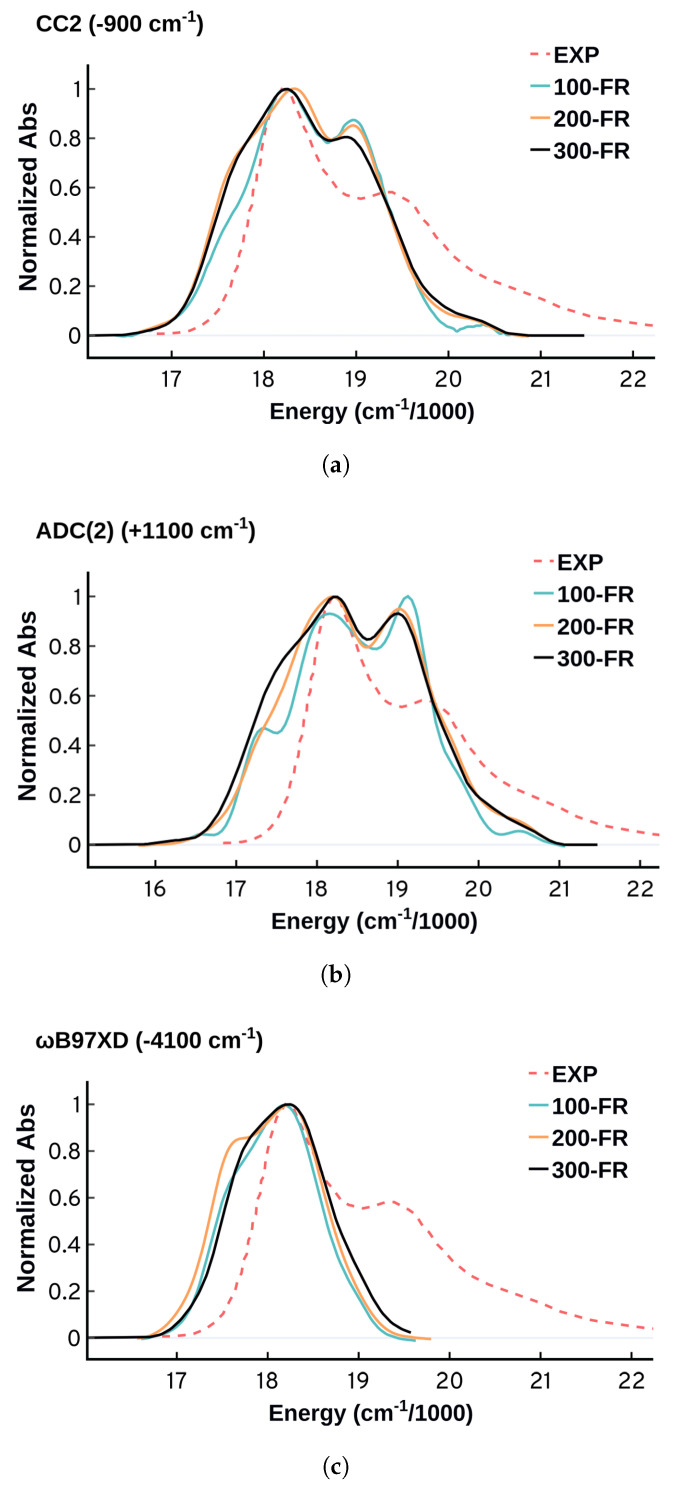
Comparing the spectra as a function of the number of configurations sampled (100 frames, 200 frames, 300 frames) obtained using (**a**) CC2, (**b**) ADC(2), and (**c**) ωB97XD. All the spectra have been normalized and shifted by the same shift as that used for the spectra obtained from 300 configurations to match the experimental maximum. The shift value and direction is included in parentheses in the charts titles.

**Figure 5 molecules-27-04062-f005:**
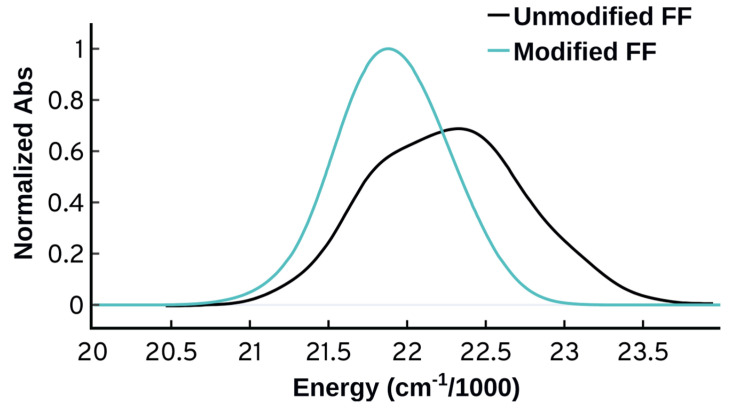
Comparison of the spectra obtained from the unmodified force field [46] (unmodified FF) and after adjusting the polymethine bond length using parameters obtained from ωB97XD/Def2SVP Cy3 optimized structure. Spectra are normalized relative to the modified one.

**Figure 6 molecules-27-04062-f006:**
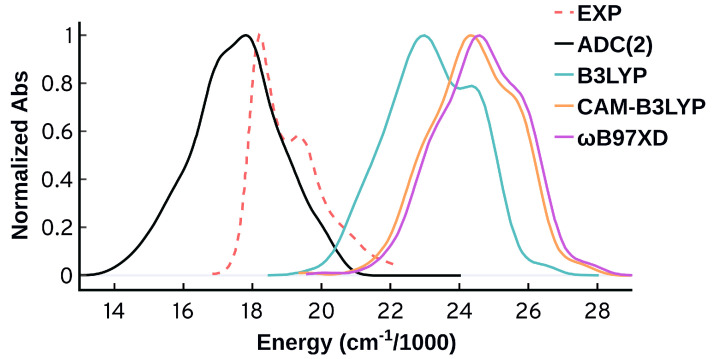
Normalized absorption spectra from an ensemble of 200 geometries generated using Wigner distribution and the harmonic approximation.

**Figure 7 molecules-27-04062-f007:**
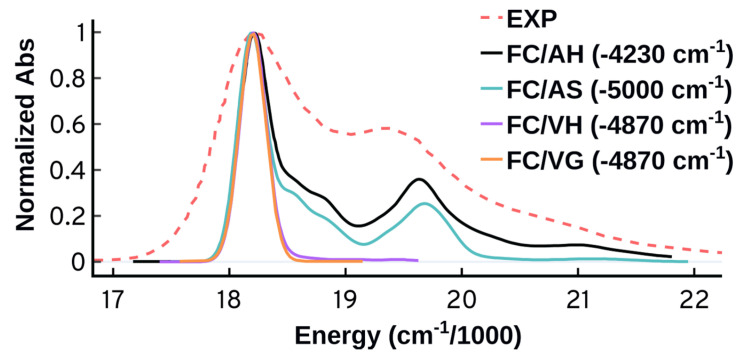
Spectra at 0 K obtained using the Franck–Condon adiabatic (AH and AS) and vertical (VH and VG) approaches. The time-independent (TI) approach at 0 K was used. The spectra are normalized and shifted to match the maximum of the experimental peak.

**Figure 8 molecules-27-04062-f008:**
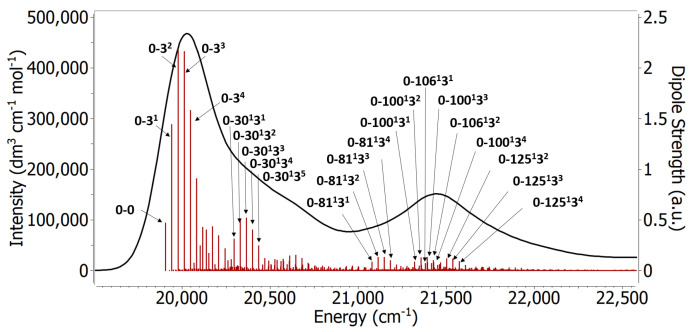
The vibronic spectrum obtained from structures optimized with ωB97XD/Def2SVP in gas phase at 0 K using the AH approach. The red sticks correspond to the vibrational excitations. The labels on the sticks are selected transitions.

**Figure 9 molecules-27-04062-f009:**
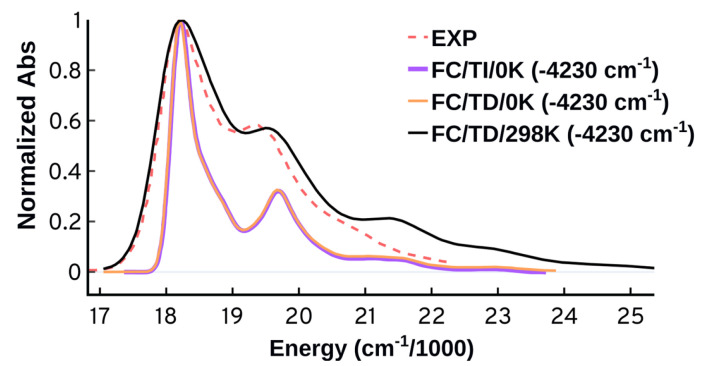
Temperature effect in the time-independent (TI) and time dependent (TD) schemes. The adiabatic Hessian (AH) approach at 0 K and 298 K was used along with structures optimized using ωB97XD/Def2SVP in the gas-phase. The spectra are normalized and shifted to match the maximum of the experimental peak.

**Figure 10 molecules-27-04062-f010:**
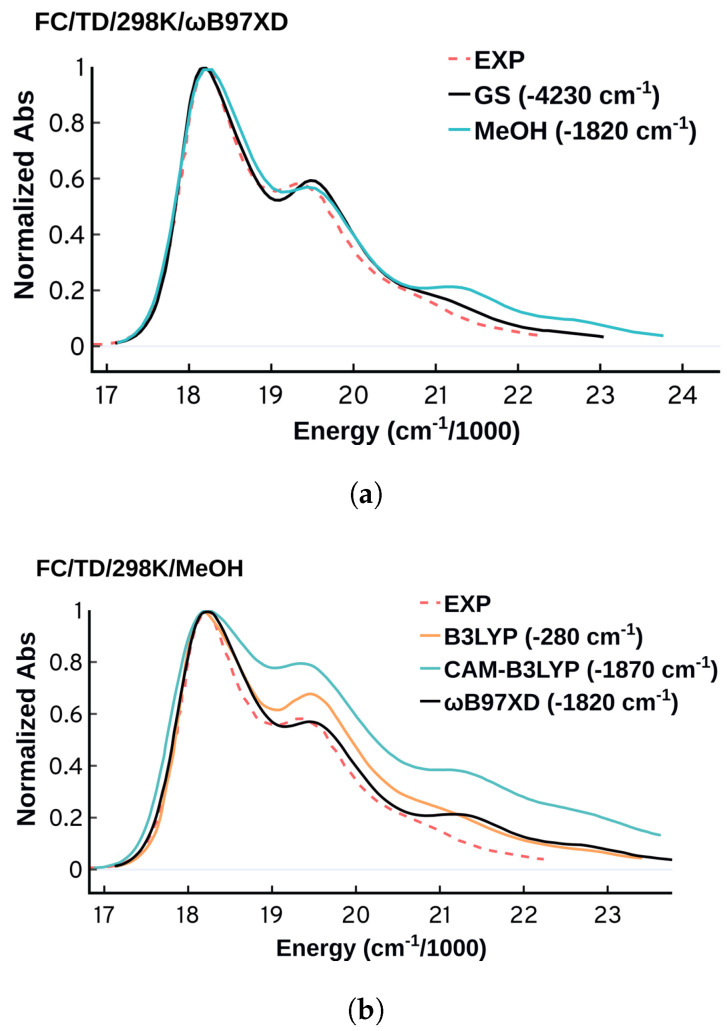
(**a**) FC spectra obtained from ωB97XD in gas-phase (ground and excited state optimized in the gas-phase) and methanol/MeOH (ground and excited state optimized in implicit solvent methanol via PCM) (**b**) Comparing the performance of B3LYP, CAM-B3LYP, and ωB97XD using methanol as an implicit solvent. All the spectra were obtained at 298 K using TD and were plotted using standard deviation of 150 cm${}^{-1}$.

**Figure 11 molecules-27-04062-f011:**
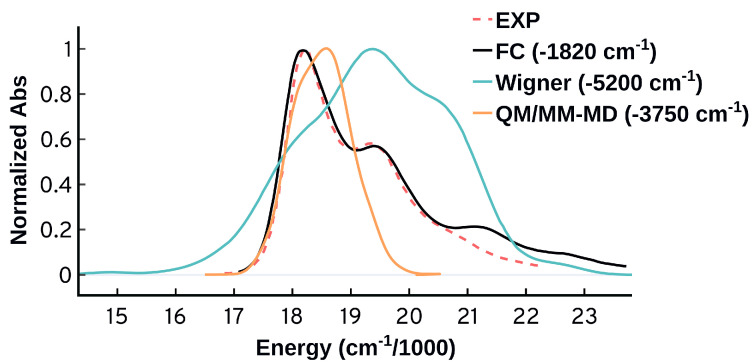
Comparing the spectra obtained from the Franck–Condon (FC), quantum sampling (Wigner), and classical sampling (QM/MM-MD) using ωB97XD/Def2SVP. The spectra here are shifted so that the first peak is matching the first experimental peak.

**Table 1 molecules-27-04062-t001:** Comparison of the polymethine equilibrium bond lengths ($R_{0}$) in Å obtained from: the equilibrium structures of the ground (ωB97XD$\left(S_{0}\right)$) and the first excited (ωB97XD$\left(S_{1}\right)$) state optimized using ωB97XD/Def2SVP; and the equilibrium bonds from Amber99SB-dyes force field (FF) [46]. *k*/FF is the force constant in kJ mol${}^{-1}$ nm${}^{-2}$ obtained from Amber99SB-dyes force field (FF) [46]. The polymethine bonds are defined in Figure 1. The newly used equilibrium C-C bond length is 1.3970 Å and the force constant is 389,275 kJ mol${}^{-1}$ nm${}^{-2}$. These values correspond to cag-cag bond parameters as implemented in the Amber99SB-dyes force field [46]. The newly used equilibrium N-C bond length is 1.3550 Å and the force constant is 387,100 kJ mol${}^{-1}$ nm${}^{-2}$, which corresponds to c2g-nhg [46] bond parameters. cag, c2g, and nhg are the atom-type [46]. The FF parameters are adapted with permission from Ref. [46]. Copyright 2014 American Chemical Society.

	$R_{0}$/ωB97XD($S_{0}$)	$R_{0}$/ωB97XD($S_{1}$)	$R_{0}$/FF [46]	*k*/FF [46]
R(N${}_{1}$,1)	1.3435	1.3712	1.391	343760
R(1,2)	1.3966	1.3960	1.4510	326770
R(2,3)	1.3938	1.4071	1.3379	470620
R(3,4)	1.3938	1.4071	1.4511	326770
R(4,5)	1.3966	1.3960	1.3390	469030
R(5,N${}_{2}$)	1.3435	1.3712	1.355	387100

## Data Availability

The data used for this study can be requested from the correspondence author.

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
