# Peer review of "Modeling the Electronic Absorption Spectra of the Indocarbocyanine Cy3"

_molecules, 2022, doi:10.3390/molecules27134062_

Round 1
Reviewer 1 Report
Report on the manuscript titled "Modeling the Electronic Absorption Spectra of the Indocarbocyanine Cy3" by Mohammed I. Sorour, Andrew H. Marcus, and Spiridoula Matsika
The authors present a thorough analysis of four different methods to compute the electronic spectrum of Indocarbocyanine Cy3. The test molecule is well chosen for such a comparison, as only a single excited electronic state contributes to its signal. The principal features of the four theoretical approaches, the classical vertical excitation at the potential minimum, the ensemble approaches with both classical and quantum sampling, and the Franck-Condon approach, are well described in the introduction and analysed in depth in the main body of the manuscript.
Some critical comments:
1) Figure 1 is amazing, it shows how very different the computed spectra are, and that none of the theoretical methods really agrees with experiment. Significant shifts of the position of the principal signal of up to 4000 cm-1 can be noticed, some methods do not reproduce the shoulder of the right-hand side of the principal signal, while some others produce the shoulder at the wrong side. These features are then carefully analysed in the manuscript.
However, not only the positions but also the intensities of computed spectra are important and depend on the computational method. The signals in Figure 1 are normalised, but for a full appreciation the authors should show the non-normalised spectra as well and draw conclusions.
2) One should be fair enough to say that if one is interested in the vibrational structure of the signal, arriving from vibrations in the upper electronic potential, then the vertical excitation and the ensemble approaches that sample ground-state configurations only, are not suitable from the beginning. Not surprisingly, the Franck-Condon approach is the method of choice.
3) It would also be helpful to discuss the case of bound-to-continuum transitions somewhere in the manuscript. For such transitions, the Franck-Condon method cannot be applied, and the ensemble approach with Wigner quantum sampling will do the job.
4) I am impressed to see the final Figure 11, which of course reflects my comments in 2). However, the authors have not made a true theoretical prediction, as the performance of a particular DFT functional cannot be predicted. It is only known after comparison with experiment. An energy shift needs to be applied, and the intensities are scaled, which is common practice though. We can at most "reproduce" an experimental spectrum, which can, of course, then be analysed in depth, such as in Figure 8, which is despite all criticism, quite impressive.
5) Can the authors suggest what method to use to produce a reliable spectrum for a compound for which experimental data do not exist. Does one need to benchmark with similar molecules? Finally, the intensity problem should be addressed.
In summary, nice and very useful work to guide future researchers. I recommend publication once the authors have addressed the points raised above.
Reviewer 2 Report
In this manuscript, the authors studied the spectral shape and the vibronic signature of the Cy3 absorption spectrum by using first-principles calculations. They tested the performance of the ensemble approach represented by the classical dynamics and quantum sampling via Wigner distribution, as well as the Franck-Condon approach and its harmonic approximations. The results showed that a reasonable absorption spectrum of Cy3 can be obtained by using the adiabatic approaches (FC/AH, FC/AS). However, revisions need to be made concerning the following issues before acceptance.
In this article, the authors discussed the influence of solvent on the spectrum of Cy3 in Figure 3, 10(a), S9 and S10. In Figure 3, they attributed the peak at 18000 cm-1 to the effect of solvent. However, if it was due to the solvent, why it appeared in the spectrum of the gas phase calculated using the FC method? Combined with the relevant Figures above, more discussions should be addressed to the influence of the used various functionals for the calculations of the solvent model.
Round 2
Reviewer 1 Report
The authors have incorporated in their revised manuscript the points raised by the two referees. I am in favour of publication.